# Oral Food Challenge in Children with Tree Nut and Peanut Allergy: The Predictive Value of Diagnostic Tests

**DOI:** 10.3390/diagnostics14182069

**Published:** 2024-09-19

**Authors:** Ludovica Cela, Alessandro Gravina, Antonio Semeraro, Francesca Pastore, Rebecca Morelli, Lavinia Marchetti, Giulia Brindisi, Francesca Olivero, Maria Grazia Piccioni, Anna Maria Zicari, Caterina Anania

**Affiliations:** 1Department of Maternal Infantile and Urological Science, Sapienza University of Rome, 00161 Rome, Italy; ludovica.cela@uniroma1.it (L.C.); alessandro.gravina@uniroma1.it (A.G.); antonio.semeraro@uniroma1.it (A.S.); f.pastore@uniroma1.it (F.P.); rebecca.morelli@uniroma1.it (R.M.); lavinia.marchetti@uniroma1.it (L.M.) giulia.brindisi@uniroma1.it (G.B.); mariagrazia.piccioni@uniroma1.it (M.G.P.); annamaria.zicari@uniroma1.it (A.M.Z.); 2Independent Researcher, 00161 Rome, Italy; francescaolivero31@gmail.com

**Keywords:** oral food challenge, food allergy, hazelnut, nut, peanut, children, pediatric

## Abstract

Food allergy (FA) affects approximately 6–8% of young children, with a peak prevalence at approximately one year of age. Tree nut and peanut allergies are among the main causes of anaphylaxis in the world. The gold standard for the diagnosis of FAs is the oral food challenge (OFC). Other diagnostic tests used in the clinical practice are skin prick tests (SPTs) and laboratory tests to measure out the presence of serum specific IgE (sIgE). In this narrative review, we collect the current evidence of the predictive value (PV) of SPTs and sIgE for the outcome of the OFCs. In literature, data are conflicting as to whether increasing sIgE concentration and wheal size in SPTs correlate with OFC outcomes. Most studies included in our review have shown that in vivo and in vitro tests may predict OFC outcomes with variable PV, but data are not conclusive; therefore, the OFC currently remains the gold standard for FA diagnosis.

## 1. Introduction

The term “nut allergy” refers to allergy to hazelnuts, almonds, cashew nuts, macadamia nuts, chestnuts, pistachios, pecans and walnuts. It is a permanent and possibly life-threatening condition; therefore, accurate diagnosis and management are crucial. The evaluation of an individual with a potential IgE-mediated food allergy (FA) includes a combination of some diagnostic tools, such as history and clinical examination, skin prick tests (SPTs), in vitro testing such as serum specific IgE (sIgE) tests and component-resolved diagnosis (CRD) and oral food challenge (OFC) [1,2,3,4,5]. At present, the gold standard for the diagnosis of FA remains OFC [6]. Although OFC is used in common clinical practice, it is troublesome, requiring an appropriate setting with a specialized team because it may cause a severe allergic reaction, up to anaphylaxis [7]. Improving the diagnostic efficacy of SPTs and laboratory tests, such as sIgE tests and CRD, might reduce the need for OFC, allowing it to be avoided when likely to be unnecessary, which would facilitate clinical practice. In the literature, data are conflicting as to whether increasing concentrations of sIgE and dimensions of the wheal in SPTs correlate with the severity of the reaction during a challenge [8,9,10,11,12]. The purpose of this narrative review is to investigate the current evidence about the predictive value (PV) of SPTs and sIgE for the outcome of OFC.

## 2. Nut Allergy

FA has a worldwide prevalence of approximately 6–8% in the pediatric population with a peak incidence at approximately one year of age. This condition has significant repercussions on the quality of life of patients and their families [13,14,15,16,17,18]. The prevalence of nut allergy is variable worldwide. In westernized countries such as the United Kingdom, Australia, the United States and Canada, the prevalence is higher than in other countries, at approximately 1 to 2% [19,20,21,22,23,24]. In Italy, nut allergy is the second leading cause of food anaphylaxis [25]. FA diagnosis is based on medical history collection and physical examination. Once the culprit food has been identified, SPTs and sIgE to the suspected food are the first-line laboratory tests performed, followed by molecular analysis of the involved allergens (CRD) [1,2,3,26]. OFC confirms the diagnosis when the history and laboratory tests performed are not conclusive, and it is also used in many allergy centers to test for tolerance acquisition [27]. Once a diagnosis is established with certainty, the standard treatment for all types of FA remains an elimination diet free of the responsible allergen [4,28,29,30,31]. Tolerance can be worked toward through oral allergen-specific immunotherapy (OIT) that, by means of continuous administration of increasing doses of the culprit food, has the ambitious goal of curing FA or at least raising the threshold of reaction. This is very important, especially in case of an accidental exposure to the culprit food [32].

Considering the wide variety of foods related to nuts and peanuts, it is important to underline the cross-reactivity concept. Cross-reactivity takes places when a patient presents allergic symptoms after the ingestion of closely related foods [33]. Among tree nut allergens, the observed cross-reactivity is high between cashews and pistachios, as well as between walnuts and pecans [34].

Furthermore, in addition to cross-reaction among nut allergens, pollen sequence homologies are often present. This phenomenon is known as oral allergy syndrome (OAS), which occurs when an individual is initially sensitized to pollen and then presents a cross-reaction to a food. These reactions are caused by the so-called pan-allergens and are responsible for non-serious symptoms often localized to the oropharynx and mouth, which hardly ever result in anaphylaxis [35].

For example, the birch pollen allergen Bet v1 is a ribosome-inactivating protein (PR-10), one of the major pan-allergens in OAS. Walnuts (Jug r5), hazelnuts (Cor a1) and peanuts (Ara h8) present homologous proteins, showing cross-reactivity with Bet v1 [36].

Another pan-allergen is the peach lipid transfer protein, Pru p3, which is the primary sensitizing allergen for cross-reactivity to other lipid transfer proteins in foods such as peanuts (Ara h 9), hazelnuts (Cor a 8), walnuts (Jug r 3) and almonds (Pru du 3) [37].

## 3. Diagnostic Tests

The diagnosis of IgE-mediated FA involves, after a thorough medical history, several diagnostic tests aimed at demonstrating IgE sensitization to a specific food allergen [4,6,38]. The SPT is a minimally invasive and safe test that can test multiple allergens in 15 to 20 min (inhalants, foods, some drugs etc.) [39,40]. Positive SPTs could indicate IgE-mediated sensitization to a certain food, and they are not necessarily indicative of FA [4,40]. In the evaluation of peanut allergy, the conventional positive results (SPT ≥ 3 mm) have poor specificity for clinical FA [41], the same holds for hazelnut allergy, where an SPT wheal diameter of 3–7 mm can be considered inconclusive [42]. When SPTs cannot be performed (for example, due to use of antihistamines, major atopic dermatitis or dermatographism), sIgE analysis can be performed. sIgE levels to food allergens can be useful; results are given in kU/L, with a positive cutoff point of 0.35 kU/L [43]. Modern research tools include molecular diagnostic techniques, such as CRD, based on the measurement of allergen-specific IgE levels. It is crucial to make a distinction between primary and secondary allergen sensitization: the first one implies the exposure to the allergen itself, while the second one affects cross-reactions to epitopes with similar structures [44]. sIgE can be directed against complete allergen extracts (which contain a mixture of proteins) or towards individual components of the suspected allergen. In this case, we use CRD, which is capable of differentiating true sensitizations from cross-reactions. It has proven to be more specific compared to the detection of sIgE against whole allergenic extracts [38]. The PV of sIgE varies based on the population studied and the specific allergen. It can be stated that undetectable sIgE levels are associated with a low risk of reaction, while higher levels increase the likelihood of an allergic reaction [45]. Regarding nut allergies, a recent systematic review has shown that the presence of specific sIgE to definite components of the allergen, such as Ara h 2 for peanuts, Cor a 14 for hazelnuts, Ana o 3 for cashews and Jug r 1 for walnuts, has high specificity in supporting the diagnosis of FA. [46,47]. Regarding almond allergy, eight native almond allergens have been characterized according to their biochemical function. However, only four of them are included in the WHO–IUIS list of allergens: Pru du 3, Pru du 4, Pru du 5 and Pru du 6. The main allergen of the almond is the storage protein Pru du 6, also known as almond major protein (AMP) or amandine, a highly thermostable protein related to severe reactions to almonds upon ingestion [48,49]. Regarding peanut allergy, 11 peanut allergens have been studied. Seed storage proteins and oleosins are the primary food allergens thanks to their high digestive and thermal stability. Among these, the main allergens involved in primary peanut allergy are Ara h 1, Ara h 2 and Ara h 3 [44]. However, some studies report cases of systemic reactions to peanuts in patients sensitized to Ara h 6, even though they do not have IgE antibodies to Ara h 1, 2, or 3 [50]. In a retrospective study conducted by Martinet et al. in 2016, they demonstrated that Ara h 2 had the best negative and positive predictive value. Moreover, the study showed that Ara h 2 titers can predict the risk of anaphylaxis [51]. Regarding cashews, the allergens identified thus far are as follows: Ana o 1 (7S globulin, vicilin), Ana o 2 (11S globulin, legumin), Ana o 3 (2S albumin) and Ana o 4 (profilin). Allergic reactions to cashews are mainly linked to the presence of IgE antibodies targeting Ana o 2 and Ana o 3. In particular, Ana o 3 is notably resistant to heat and digestion, and it is associated with the occurrence of severe allergic reactions [48]. Regarding hazelnut allergy, the following are the main allergens recognized: Cor a 1 (protein PR-10), Cor a 2 (profilin), Cor a 6 (isoflavone reductase approval), Cor a 8 (non-specific LTP type 1), Cor a 9 (11S legumin, storage protein), Cor (LBP, luminal binding protein), Cor a 11 (7S vicilin, storage protein), Cor a 12 (oleosin of 17 kDa molecular weight), Cor a 13 (oleosin of molecular weight 14–16 kDa) and Cor a 14 (2S albumin) [48]. Cor a 1 and Cor a 2 are homologues of Bet v 1; therefore, they are responsible for cross-reactive IgE between pollen and plant-derived foods. Moreover, Cor a 1 is rather unstable to heating and digestion; therefore, it has been mainly implicated in OAS in response to hazelnuts. Cor a 8, Cor a 9 and Cor a 11 are the main allergens implicated in severe allergic reactions to hazelnuts, but patients with these allergies are generally sensitive to more than one recognized allergen [52]. Chestnut allergy is the third most common FA in adults worldwide. The allergens characterized to date are Cas s 1 (PR-10, Bet v 1 homolog), Cas s 5 (chitinase), Cas s 8 (non-specific LTP 1) and Cas s 9 (small thermal shock protein class I). Among them, the main allergens used as diagnostic tools in people allergic to chestnuts are Cas s 5 and Cas s 8. [48]. However, chestnut allergy is rarely isolated, e.g., Cas a 5 is associated with latex–fruit syndrome, indicating significant cross-reactivity with rubber latex hevein [53]. For walnut allergy, the allergens identified are as follows: Jug r 1 (2S albumin, spare protein), Jug r 2 (7S vicilin, reserve protein), Jug r 3 and Jug r 8 (non-specific lipid transporting protein (nsLTP)), Jug r 4 (11S globulin, spare protein), Jug r 5 (PR-10), Jug r 6 (vicilin) and Jug r 7 (profilin). IgEs specific to Jug r 1 were detected in most walnut-allergic patients in a US-based study, while sensitization to Jug r 3 seems to be prevalent in Italian patients. [54]. In pecan allergy, three pecan allergens have been characterized thus far: Car i 1 (2S albumin), Car i 2 (vicilin) and Car i 4 (11S globulin) [55]. Proteins resistant to high temperature and digestion are seemingly involved in severe systemic reactions to pecan, but protein contact eczema was also described, as in a case described in 2006 by Joyce et al., where a vesicular cutaneous reaction occurred in a patient after a prolonged contact with pecans, reminding that prolonged contact with allergenic food can lead to hypersensitivity reactions [48,56]. Regarding pistachio allergy, the following allergens have been identified: Pis v 1 (2S albumin), Pis v 2 (11S globulin), Pis v 3 (vicilin) and Pis v 4 (peroxide dismutase). However, allergy to pistachios is considered relatively uncommon. When it does occur, allergic symptoms can include oral cavity reactions, skin issues, angioedema and severe anaphylaxis [48]. The macadamia nut allergens have not been defined yet [48].

The main steps for the diagnosis of nut allergy are summarized in the flowchart below (Figure 1).

## 4. Oral Food Challenge

OFC is the diagnostic gold standard for FA, both IgE- and non-IgE-mediated [4,6]. It should be administered in patients with a clinical history strongly suggestive of FA but with negative or inconclusive results of SPTs or sIgE tests [3,4,38]. When we have a suspicion of multiple allergies, we should start conducting the OFC with foods that have high nutritional value or are ubiquitous, as in the case of tree nuts and peanuts, contained in traces in several foods. OFC is generally conducted by specialists in a controlled environment, in case intervention is needed. For example, subjects allergic to peanuts or tree nuts may have a higher risk of severe reaction during OFC. It should not be performed in patients with clinical history of anaphylaxis. Patients should be on minimal or no symptomatic medication, such as antihistamines and corticosteroids. The OFC can be open or a double-blind or single-blind placebo-controlled challenge. This type of test is utilized to either confirm an allergy or verify the development of tolerance. It consists of three steps: the patient ingests a small portion of the food, in its natural form, to which the allergy is suspected; the patient then needs to be constantly monitored in case of an adverse reaction; if there is no adverse reaction, the patient can be fed increasing amounts of that food until a regular portion size is reached. The goal is to identify the lowest provoking dose of the food allergen. In the event of an allergic reaction, the test must be immediately stopped; otherwise, negative results indicate oral tolerance [57,58,59,60,61,62,63]. In a retrospective review of more than 700 food challenges, 19% of patients reacted, including 2% who required epinephrine treatment [64].

## 5. Methods

The present review provides a look of the recent literature to investigate the PV of SPTs and specific sIgE for the outcome of OFC in tree nut and peanut allergies, with the aim of simplifying the diagnosis of these FAs. Evidence published from 1997 to 2023 was searched using the PubMed and Scopus databases. We excluded all duplicate articles, all reviews and meta-analyses and all articles whose titles or abstracts did not fit with our topic. The final number of articles was twenty. The keywords used for the search were “predictive value”, “oral food challenge”, “OFC”, “SPTs” “nut allergy” and “peanut allergy”.

## 6. Discussion

We collected the evidence already available in the literature about the PV of sIgE and SPTs for the outcomes of OFC in children with tree nut and peanut allergy. Although the data are not conclusive, the results are encouraging, and we have drawn conclusions that can be useful in clinical practice. We discuss the details in the table below (Table 1).

## 7. Observational Studies

The few published observational studies all confirm that the size of the SPT wheal, (which is the best predictor) and specific IgE levels are related to the risk of allergic reaction.

In detail, the study conducted by Elegbede et al. in 2019 developed a model for the threshold dose distribution curve based on routinely collected data, such as SPT results and Ara h 2 sIgE levels. In their study, the wheal size of SPTs and the value of Ara h 2 sIgE correlated with the risk of peanut allergic reactions, with a beta coefficient of 0.05 [0.02; 0.08] associated with the risk; the 95% CI was 0.01 [0.01; 0.02]. Moreover, according to this study, the Cox model is the most effective statistical model for predicting threshold doses, based on some variables. They concluded that a lower threshold dose of reaction is often observed in female subjects, in the absence of atopic dermatitis, when there is a high level of Ara h 2 sIgE and when there is a larger SPT wheal size [76].

A single-center, cross-sectional study published by Chua et al. in 2021 determined the ideal cutoff values for sIgE level, SPT wheal size and CRD in Chinese children based on outcomes of peanut OFCs. They performed SPTs, blood tests for peanut sIgE and CRD (Ara h 1, 2, 3, 8 and 9 sIgE) and OFC on all patients, and only SPTs reached statistical significance (AUC 0.91, *p* = 0.0001); therefore, they added retrospective data from seven patients to optimize the power. For this new study population, the ROC curves of both SPTs and Ara h 2 sIgE achieved statistical significance (AUC 0.9, *p* < 0.0001, and AUC 0.72, *p* = 0.02, respectively). The Chinese group demonstrated that SPTs are the best predictor of peanut allergy in the Chinese pediatric population, followed by Ara h 2 sIgE, and that patients with wheal size ≥ 6 mm on an SPT have a high probability of being allergic to peanuts; therefore, in these patients, OFC should be avoided or considered with caution [80].

## 8. Retrospective Studies

Despite the many published retrospective studies, there is no univocal indication of a biomarker that can be useful for predicting a positive outcome of a food challenge, which remains the only viable way to evaluate tolerance.

In detail, to establish the role of the CAP System FEIA (fluorescence enzyme immunoassay) in diagnosing IgE-mediated FA, Sampson et al. conducted a retrospective study in 1997, analyzing sera from 196 patients (0.6–17.9 years old) for sIgE to eggs, milk, peanuts, soy, wheat and fish. Comparing these data with OFC results and convincing histories of allergic reactions, a cutoff value set to 0.35 kUA/L for egg-, milk-, peanut- and fish-specific IgEs showed excellent sensitivity (ranging from 94% to 100%) and high negative predictive accuracy (>97%) in a supposed population of standardized patients where the prevalence of specific FA is 10%. These results were comparable to those found with SPTs; in contrast, the specificities (25–65%) and corrected positive predictive accuracies were very poor, even lower than the SPTs’ values. Furthermore, they calculated values corresponding to 95% and 90% positive and negative predictive accuracies, suggesting that patients with food sIgE levels greater than the 95% PV (which is ≥15 kUA/L for peanut) do not need an OFC to confirm the diagnosis, considering that they are extremely likely to develop a clinical reaction during the challenge [65].

OFC safety and the PV of the SPT in peanut allergy were assessed in 2001 in a retrospective study conducted by Pucar et al. They performed OFCs on the study population; all children with SPT < 5 mm had a negative challenge, and all children with a suggestive history and SPT wheal size ≥ 5 mm had a positive OFC. They concluded that OFC is in general necessary to support the diagnosis when the allergic symptoms are suggestive and the SPT is borderline positive, i.e., 3–4 mm. The NPV and sensitivity of SPTs are excellent; therefore, in patients with negative SPTs, OFC is usually negative [66]. 

In 2002, Kagan et al. conducted a retrospective study on 47 patients who had a positive peanut SPT with no previous oral consumption of peanuts and had undergone a peanut OFC. Overall, 49% of the OFCs performed were positive, with the mean SPT wheal size (95% CI) in children having a positive and negative OFC being 10.3 mm (CI, 8.9 to 11.8) and 6.3 mm (CI, 5.3 to 7.3), respectively. At SPT = 5 mm, the sensitivity, specificity, NPV and PPV (95% CI) were 100% (85.2 to 100), 12.5% (2.7 to 32.4), 100% (29.2 to 100) and 52.3% (36.7 to 67.5), respectively. If the cutoff of the SPTs was considered to be 5 mm, the sensitivity and NPV were 100%, but the sample population was too small to draw conclusions about the PV of SPTs and about the role of sIgE [67].

To evaluate the PV of sIgE, Perry et al. conducted a retrospective study in 2004, using 604 OFCs performed on 391 pediatric patients. The analyzed foods were eggs, milk, peanuts, wheat and soy. A total of 173 OFCs were performed using peanuts; among these, 59% of patients passed, with a median value of sIgE in negative and positive OFC of 0.5 kUA/L and 1.9 kUA/L, respectively (*p* < 0.001). For patients with a well-documented previous peanut reaction, the trend of an increasing failure rate correlated with higher peanut sIgE (*p* < 0.01). Overall, 76% of patients with a negative OFC had sIgE levels < 0.35 kUA/L, while no one who passed the OFC had a sIgE level > 5 kUA/L. For those patients without a clear reaction history, 88% of patients with peanut sIgE levels < 0.35 kUA/L had a negative OFC, and 77% of patients with peanut sIgE levels > 5 kUA/L passed the OFC as well. Median peanut sIgE levels were statistically different in patients with positive and negative OFC. The study recommends that patients with a clear history of reaction to peanuts should undergo OFC when the sIgE level is <2 kUA/L, whereas a cutoff level of 5 kUA/L is recommended for those without a clear clinical history of reaction [68]. 

A retrospective study conducted in 2007 by Ueno et al., involving 51 patients who underwent 87 OFCs, showed that sIgE titers do not differ significantly between patients who pass the OFC and those who do not. The skin index (which is calculated as the ratio of the diameter of the wheal induced by the allergen to that induced by histamine) and SPT results were significantly different between the two groups (*p* < 0.01 and *p* = 0.03, respectively). Thus, the authors suggested that the skin index may be helpful for predicting a positive outcome of food challenge [69].

Van Veen et al. tried to assess the correlation between peanut allergy and peanut sIgE levels in peanut-sensitized children, as well as how factors such as the clinical setting, asthma and eczema influenced this relationship. At the beginning of the study, 427 children were enrolled (all sensitized), but only 280 were assessed in the study. The studied population was divided into three groups: allergy (*n* = 52, based on both previous OFC and history); possible allergy (*n* = 38, with only subjective symptoms, a lack of clearly reproducible symptoms or no ingestion of relevant doses of peanuts); and no allergy (*n* = 190). Peanut sIgE levels were higher in allergic patients than in those who were not allergic (*p* < 0.001). The group with possible peanut allergy was not included in further analyses of this association. The peak likelihood ratio of a positive peanut sIgE test indicating peanut allergy was 16.3 (sensitivity 42%, specificity 97%, PPV 79%, NPV 86%) at a sIgE level of 30.0 kU/L. Conversely, the lowest likelihood ratio for a negative test was 0.2 (sensitivity 96%, specificity 15%, PPV 24%, NPV 79%) at a sIgE level of 0.6 kU/L. Their retrospective study, published in 2013, showed that a history of eczema could influence the relationship between peanut sIgE and peanut allergy. However, the proportion of peanut-sensitized participants who were defined as having peanut allergy was smaller, and the PV of peanut sIgE levels for clinical peanut allergy was weaker than in other studies. Moreover, peanut allergy prediction did not achieve a probability of 95% even at the highest level of peanut sIgE (>100 kU/L) [70]. 

The correlation of basophil activation and sIgE with OFC outcome and severity of the reaction has been analyzed by Song et al. in 2015 regarding peanut and tree nut allergy. The median SPT wheal sizes were significantly different between positive and negative OFC (10 mm vs. 5.5 mm; *p* < 0.001). Moreover, there was a significant difference in the median sIgE levels in subjects with positive vs negative OFC (26.9 kUA/L vs 2 kUA/L, *p* < 0.001). The total IgE (tIgE) levels, the sIgE/tIgE ratios and the allergen-specific IgG4 levels were not different between the two groups, but the sIgE/sIgG4 ratios clearly discriminated positive and negative reactions (*p* < 0.05). Moreover, a weak positive correlation was found between sIgE to the peanut component Ara h 2 and OFC severity scores (*p* = 0.038), while there was a weak negative correlation between IgE to Ara h 8 and OFC severity scores (*p* = 0.0139). Therefore, the authors concluded that the best predictor of OFC outcome was the basophil activation test (BAT) at the highest dose (200 ng/mL) (*p* < 0.001) [73].

A retrospective, international, multicenter case record was published in 2018 by Arkwright et al. All patients were tested for peanut SPT, peanut sIgE and peanut Ara h 2, and they all underwent OFC to peanuts. They demonstrated that the wheal size on SPTs (but not peanut sIgE) correlated with the risk of anaphylaxis (relative risk 1.2, 95% CI: 1.1–1.3) [75]. 

In a 2019 retrospective study, McWilliam et al. tried to evaluate the SPT wheal size that correlated with 95% PPV to a positive OFC for cashews in both a population and a clinic-based cohort of children. Cashew OFCs were positive in 19.9% of patients in the clinic cohort and in 62.1% of patients in the population cohort. The authors reported a 95% PPV for a positive OFC for cashews with an SPT wheal size of 10 mm in the population cohort and 14 mm in the clinic cohort, with variability depending on the population characteristics [77].

In 2019, Virkud et al. found that non-tolerance in almond OFCs was linked to higher almond sIgE levels (*p* < 0.001), larger SPT wheal size (*p* = 0.001) and higher peanut IgE levels (*p* = 0.003). Despite this correlation, the estimated 95% predicted probability of a positive OFC occurs at a SPT wheal size of 46 mm and sIgE of 174 kU/L. These values were significantly higher than those observed in the cohort. Most failed OFCs caused mild reactions, with only 0.5% of patients having anaphylaxis. The results suggest that almond OFCs are relatively safe, with the potential for some to be conducted at home or with higher patient-to-staff ratios in clinical settings [78].

Chong et al. found out that larger SPT wheal sizes and sIgE levels are associated with an increased likelihood of a clinical peanut allergy. In particular, a wheal size ≥ 8 mm and peanut sIgE ≥ 6 kU/L each provided a 95% PPV for clinical reaction to peanuts. They suggested that these findings could also provide a basis for choosing which peanut-sensitized patients should undergo an OFC [79]. 

A study by Lee et al. published in 2021 tried to determine the ability to detect sIgE antibodies against walnuts using self-reported FA symptoms. Based on clinical history, they designed five classes: Class 1 (consistent clinical history, direct isolated intake, anaphylaxis or hives), Class 2a (class 1 with inconsistent clinical history), Class 2b (anaphylaxis or hives following indirect or mixed intake), Class 2c (consistent clinical history, direct or indirect/isolated or mixed intake, itching, vomiting or diarrhea), Class 3 (class 2c with inconsistent clinical history or no reaction to direct, isolated exposure). Regarding walnut allergy, sIgE sensitization rate in class 1 cases was high (90%) and the most predictable with an accuracy rate of 76.5%. This study suggests that a detailed medical history could screen for potential tree nut allergy and for IgE sensitization, helping to define the necessity of further OFC testing [81]. 

Kubota et al. investigated the utility of macadamia nut sIgE for predicting anaphylaxis in macadamia nut allergy. The study population (*n* = 41) was divided into an allergic group (MdA, *n* = 21) and a tolerant group (non-MdA, *n* = 20). In the MdA group, eight subjects (38%) experienced anaphylaxis (An group), while thirteen children (62%) did not (non-An group). Macadamia nut sIgE levels were significantly higher in the An group than in the non-An group (median 7.97 kUA/L vs. 1.92 kUA/L, *p* = 0.02) and non-MdA group (7.97 kUA/L vs. 1.90 kUA/L, *p* < 0.001). The optimal macadamia nut sIgE cutoff value was 3.76 kUA/L, with a sensitivity, specificity, PPV and NPV of 100.0%, 75.8%, 50.0% and 100.0%, respectively. The results of this retrospective study suggest that macadamia nut sIgE levels are positively linked to the risk of anaphylaxis; therefore, OFC should be performed cautiously in this population [82].

In 2023, Grinek et al. conducted a retrospective study to assess the prognostic ability of epitope-specific antibodies to predict the result of OFC at 5 years. Five different combinations of specific antibodies were evaluated: (1) qualitative epitope-specific IgE (sesIgE) alone; (2) sesIgE + Ara h 1, 2, 3 and 9 (“sesIgE+Ara h sIgE”); (3) sesIgE + peanut-specific IgE (“sesIgE+PN-sIgE”); (4) both sIgE to component proteins and peanut (“sesIgE+Ara h sIgE+PN-sIgE”) and (5) all the previous antibodies, including ses-IgG4 (“sesIgE+Ara h sIgE+ses-IgG4”). The performance of all the models showed that predictive performance improved with age, with reliable predictions possible starting at 1 year of age. At 1 year, “sesIgE+Ara h sIgE” resulted in the best model, with the addition of PN-sIgE showing similar results (“sesIgE+Ara h sIgE+PN-sIgE”), while the addition of ses-IgG4 values did not improve the performance. The study also evaluated how this model performed in comparison to the simple ImmunoCAP-based cutoff of 0.35 kUA/L of sIgE to peanut; at 1 year of age, the “sesIgE+Ara h sIgE” model achieved an accuracy of 83% [82%−85%], significantly superior (*p* < 0.001) to the performance of the PN-sIgE-based method (accuracy of 76% [74%−77%]). In conclusion, sesIgE profiles obtained in high-risk children avoiding peanuts at about 1 year of age showed promising performance as prognostic biomarkers for peanut OFC outcome at 5 years of age. Models combining sesIgE and sIgE to component proteins performed best at predicting allergy status at 5-year OFC, while adding IgE to peanut extract did not improve performance [83]. 

Koutlas et al., who analyzed 663 OFCs conducted on 510 pediatric patients, investigated the association of SPT wheal size and sIgE with OFC outcomes. The OFC pass rate was 84% for peanuts, 86% for cashews, 83% for walnuts and it 100% for almonds and Brazil nuts. Regardless of the trigger, a history of anaphylaxis was correlated with a higher failure rate (22% vs. 11.6% [*p* = 0.01]; odds ratio = 2.14 [95% CI = 1.21–3.80]). This study showed that cutoffs of sIgE not exceeding 2 kU/L and SPT wheal size not exceeding 5 mm result in a failure rate of approximately 13% for the most frequent allergens (nonbaked milk, nonbaked eggs and nuts) [84].

## 9. Prospective Studies

Prospective studies have investigated the role of specific IgE cutoffs; however, despite the identification of specific cutoffs relating to the individual populations examined in each study, standardization has not yet been achieved.

In a 2014 prospective study, Gupta et al. tried to assess the accuracy of the sIgE/tIgE ratio in predicting the outcomes of OFCs with peanuts, tree nuts and other foods. The ratio was higher in patients who failed the OFC than in those who passed the challenge (failed 1.48% vs. passed 0.49%). Moreover, ROC curves showed the significant accuracy of the ratio, compared to sIgE alone, in predicting outcomes of OFCs, and it applied especially to peanuts (*p* = 0.08) and tree nuts (*p* = 0.14) [71].

Beyer et al., in 2015, tried to identify cutoff levels of sIgE to peanut and hazelnut components for the prediction of OFC. In that study, levels of 14.4 kU/L and 42 kU/L of Ara h 2-sIgE indicated a 90% and 95% probability for a positive peanut challenge, respectively. Levels of 48 kU/L of Cor a 14-sIgE indicated a 90% probability of being allergic to hazelnuts. They concluded that Ara h 2- and Cor a 14-sIgE discriminate between allergic and non-allergic children more effectively than hazelnut- or peanut-sIgE and that they could be useful to estimate the probability of a positive OFC [72].

The role of the diagnostic tests and their cutoff values in hazelnut allergy were also investigated by Buyuktiryaki et al. In this study population, the median SPT wheal diameter and the levels of hazelnut-sIgE and Cor a 14-sIgE were significantly higher in allergic children than in tolerant ones (*p* = 0.004, *p* < 0.001 and *p* < 0.001). They concluded that the cutoff levels that best predict clinical reactivity to hazelnuts are sIgE 3.15 kU/L and SPT wheal diameter 7.5 mm and that Cor a 14 is a more reliable diagnostic test than others in predicting clinical reactivity [74].

## 10. Results

In this review, we have collected 20 articles from 1997 to 2023 concerning tree nut and peanut allergies and the PV of various diagnostic tests. The studies establish different cutoffs for SPTs, sIgE and CRDs, aimed at improving the accuracy of allergy diagnosis and potentially reducing the need for OFCs.

For peanut allergy, a negative SPT (<5 mm) is highly reliable for excluding the allergy [66], while a positive SPT (≥8 mm) is strongly predictive of an allergic reaction [79], making OFC often unnecessary in both cases. High sIgE levels (≥15 kUA/L) confirm peanut allergy with high certainty, potentially eliminating the need for OFC [65,79]. Additionally, even a lower sIgE threshold (≥6 kU/L) has a high PPV for clinical reactions, aiding clinical decision-making [79]. Additionally, levels of Ara h 2 (>42 kU/L) strongly predict a positive OFC, potentially eliminating the need for this challenge [72].

Regarding hazelnut allergy, cutoffs including an SPT result of 12 mm, an sIgE level of 10.2 kU/L and a Cor a 14 level of 48 kU/L are all highly predictive of the allergy, reducing the necessity of OFC [72,74].

Regarding almond allergy, an sIgE level of 174 kU/L and an SPT result of 46 mm are strongly associated with a positive clinical reaction, supporting the diagnosis and often negating the need for OFC [78].

Lastly, for macadamia nut allergy, an optimal sIgE cutoff value of 3.76 kUA/L provides 100% sensitivity and NPV, confirming the diagnosis without the need for additional invasive testing [82].

In conclusion, several cutoff thresholds for SPT, sIgE and CRD that strongly predict food allergy have been reported with variable results. Differences in study design, characteristics of enrolled patients and OFC protocols may account for these conflicting results. Overall, these findings suggest that specific cutoff values can significantly enhance diagnostic accuracy and may reduce the need for OFCs. However, it is important to note that this is not a systematic review; therefore, these values should be interpreted with caution. These thresholds currently cannot be generalized or standardized. The information provided offers a general perspective based on the studies reviewed and summarized in Table 1, rather than definitive diagnostic thresholds.

## 11. Conclusions

To date, several retrospective, observational and prospective studies have been conducted to correlate the PV of allergy diagnostic tests, such as the SPT wheal diameter, food-sIgE and CRD, with the outcomes of OFC in children with tree nut and peanut allergy. The majority of the studies included in our review have shown a correlation linking SPTs and laboratory tests with the outcome of OFCs and the severity of reactions during OFCs. The most convincing results concern data about the sensitivity and NPV of wheal size in SPTs and IgE levels: in many studies, undetectable sIgE levels and negative SPTs are associated with a low risk of reaction during OFC. Moreover, in other studies, a larger size of SPT wheal and a higher level of sIgE correlate with a higher risk of reacting to a small amount of the culprit food. 

Some of the studies included in this review analyzed patients allergic to food belonging to different categories, while other included patients who showed multiple allergies, but mostly because of cross-reacting food.

Although these results are encouraging, we must consider the differences among all these studies in terms of the type of research and the characteristics of the enrolled population. In conclusion, further studies are needed to confirm the PV of SPTs and food sIgE for the OFC outcome. Currently, the OFC remains the gold standard and the tool for FA diagnosis in the daily clinical practice of the pediatric allergist.

## Figures and Tables

**Figure 1 diagnostics-14-02069-f001:**
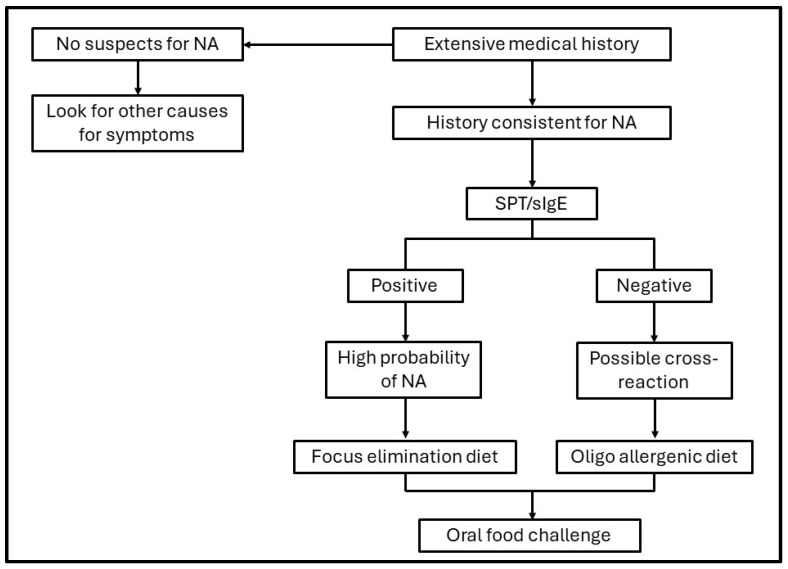
Schematic flowchart of the main steps required for the diagnosis of nut allergies. NA: nut allergy; SPT: skin prick test.

**Table 1 diagnostics-14-02069-t001:** Clinical studies assessing the relationship between OFC results and diagnostic allergy tests in children with tree nut allergy and peanut allergy.

Author, Year, Nationality	Study Design	Study Population	Food Allergy	SPT	Specific sIgE	Molecular Findings	Outcome	Results
Sampson et al., 1997, USA [65]	Retrospective	196 (0.6–17.9 y; 5.2 y median age)	Peanuts and others	Cutoff SPT = 3 mm Sensitivity = 90%, specificity = 29%, “corrected” NPV = 96%, “corrected” PPV = 12%	sIgE = 0.35 kUA/L 95% PPV = 15 kUA/L 90%PPV = 9 kUA/L 85% NPV = 0.35 kUA/L	Not studied	sIgE: sensitivity = 97%, specificity = 38%, “corrected” NPV = 99%, “corrected” PPV = 15%	Patients with peanut IgE > 15 kUA/L do not need OFC to confirm the diagnosis
Pucar et al., 2001, Canada [66]	Retrospective	140 (15 m–17 y; 5.2 y median age)	Peanuts	SPT < 5 mm = OFC negative; SPT ≥ 5 mm = OFC positive	Not studied	Not studied	Sensitivity = 100% Specificity = 62.3% PPV = 28.1% NPV = 100%	In patients with SPT < 5 mm, OFC is usually negative If SPT > 5 mm, OFC is necessary
Kagan et al., 2003, Canada [67]	Retrospective	47 (4.5–7.6 y; 5.2–6.4 median age)	Peanuts	SPT = 5 mm, sensitivity and NPV = 100%; SPT = 10 mm, PPV = 80%; SPT = 12 mm, PPV = 90%.	Median sIgE = 0.63 kUA/L in TS; 13.1 UA/L in AS	Not studied	SPT = 5 mm, sensitivity = 100%, NPV = 100%	Small sample size limits applicability of this value
Perry et al., 2004, USA [68]	Retrospective	391 (5.3 y median age), including 173 tested for peanut allergy (5.6 y median age)	Peanuts and others (milk, peanuts, wheat, soy)	Not studied	Median IgE = 0.5 kUA/L in TS IgE = 1.9 kUA/L in AS (*p* < 0.001)	Not studied	Clear history: at IgE < 0.35 kUA/L, 76% passed OFC; at IgE > 5 kUA/ L, none passed OFC No clear history: at IgE < 0.35 kUA/L, 88% passed OFC; at IgE > 5 kUA/L, 77% passed OFC	Patients with a positive history should be challenged at IgE < 2 kUA/L; patients without a positive history should be challenged at IgE = 5 kUA/L
Ueno et al., 2007, Japan [69]	Retrospective	51 (35 m median age)	Peanuts (4 OFCs) and others	SPT: sensitivity = 100%, specificity = 11%, PPV = 41%, NPV = 100%	IgE: sensitivity = 91%, specificity = 15%, PPV = 40%, NPV = 73%	Not studied	SPT results are significantly different between the two groups (*p* = 0.03); IgE is not significantly different	Skin index correlates better than SPT result with OFC outcome
Van Veen et al., 2013, Netherlands [70]	Retrospective	427 (0–18 y) 280 completed the study AP: 52 TP: 190 PPA: 38	Peanuts	Not studied	AS have higher peanut-specific IgE levels than TS (*p* < 0.001)	Not studied	Probability of positive OFC < 95% even if peanut-specific IgE >100 kU/L Eczema is most strongly related to peanut allergy (OR 3.20, 95% CI 1.30–7.93)	Weak relationship between peanut-sIgE and peanut allergy (limited utility of sIgE in diagnosis)
Gupta et al., 2014, USA [71]	Prospective	161 (11 m to 18 y; 4.0 y median age)	Peanuts, tree nuts and others	Not studied	sIgE range in the overall study: 0.1–55.7 kUA/L tIgE range: 2.2–5000 kU/L Ratio range: 0.1–10.5%	Not studied	Peanuts: OFC ratio vs. sIgE AUC = 0.78 vs. 0.56 (*p* = 0.08) Tree nuts: OFC ratio vs. sIgE AUC = 0.85 vs. 0.60 (*p* = 0.14)	Ratio showed a significantly higher accuracy than sIgE alone in predicting outcomes of OFCs involving persistent food allergens
Beyer et al., 2015, Germany [72]	Prospective multicenter	353 (2–8 y): 210 OFCs with peanuts, 143 OFCs with hazelnuts.	Peanuts and hazelnuts	Not studied	Not studied	Ara h 2 = 0.35 kU/L, sensitivity = 86%, specificity = 86%; Cor a 14 = 0.35 kU/L, sensitivity = 85%, specificity = 81%.	Ara h 2 at 14.4 kU/L and 42 kU/L indicate a 90% and 95% probability of a positive OFC, respectively Cor a 14 at 48 kU/L indicates a 90% probability of a positive OFC	Ara h 2- and Cor a 14-sIgE could be useful for estimating the probability of a positive OFC.
Song et al., 2015, USA, China [73]	Retrospective	67 (16 y median age)	Peanuts, tree nuts and others	SPT results are significantly different between positive and negative OFC (*p* < 0.001)	Median sIgE levels are significantly different between positive and negative OFC (*p*< 0.001).	Weak positive correlation between Ara h 2 and Ara h 8 OFC severity scores (*p* = 0.038 and *p* = 0.0139, respectively)	SPT wheal size, sIgE level and sIgE/sIgG4 ratio are correlated with OFC outcome (*p* < 0.001, *p* < 0.001 and *p* < 0.05)	SPT wheal size and sIgE level are useful in predicting the presence of FA but not the severity of the reaction
Buyuktiryaki et al., 2016, Turkey [74]	Prospective	64 (2.1 to 7.2 y; 3.4 y median age)	Hazelnuts	SPT = 5.25 and = 12 mm indicate a 50% and 95% probability of clinical reactivity	IgE values of 4 and 10.2 kU/L to hazelnut extract indicate a 50% and 95% probability of clinical reactivity	Cor a 14 sIgE = 0.48 and = 1.0 kU/L indicate a 50% and 95% probability of clinical reactivity	SPT, hazelnut IgE and Cor a 14 sIgE are higher in AS than in TS (*p* < 0.004, *p* < 0.001 and *p* < 0.001).	Cor a 14 is the most reliable diagnostic test for predicting clinical reactivity in children
Arkwright et al., 2018, United Kingdom [75]	Retrospective, multicenter, case record	1634 (1–18 y)	Peanuts	Sensitivity, specificity, PPV and NPV of SPT to peanuts ≥ 3 mm are 93%, 56%, 50% and 94%; those of SPT ≥ 8 mm are 47%, 94%, 81% and 79%	Sensitivity, specificity, PPV and NPV of peanut-specific IgE ≥ 0.35 AUK/L are 80%, 58%, 48% and 86%	Sensitivity, specificity, PPV and NPV of Ara h 2 IgE ≥ 0.35 AUK/L are 57%, 96%, 87% and 84%	SPT ≥ 3 mm had the best NPV (94%), and Ara h 2 ≥ 0.35 AUK/L provided the best PPV (87%)	Wheal size in SPT correlates with the risk of anaphylaxis (relative risk 1.2, 95% CI 1.1–1.3)
Elegbede et al., 2019, France, Belgium, Luxembourg [76]	Observational, multicenter	785 (2 to 27 y; 9 y median age); 204 OFCs	Peanuts	Higher size of SPT = higher risk of reacting to a small quantity of peanuts	Not studied	Higher Ara h 2 sIgE level = higher risk of reacting to a small quantity of peanuts	SPT and Ara h 2 correlate with the risk of peanut allergic reaction (beta coefficient and 95% credible interval of 0.05 and 0.01 respectively)	The Cox model is the most effective statistical model to predict threshold doses, based on gender, SPT and Ara h 2
McWilliam et al., 2019, Australia, UK [77]	Retrospective	145 (7.8 y median age) Clinical cohort: 286 (7.1 y median age)	Cashews	Population cohort: SPT > 8 mm in 91% of patients with positive OFC Clinical cohort: SPT > 8 mm in 59% of patients with positive OFC	Not studied	Not studied	Population cohort: SPT cutoff for 95% PPV is 10 mm Clinical cohort: SPT cutoff for 95% PPV is 14 mm	A larger SPT wheal size may be more suitable than the 8 mm cutoff to guide clinical decisions
Virkud et al., 2019, USA [78]	Retrospective	590 (1–66 y)	Almonds	Positive OFC is correlated with larger wheal size in the SPT (*p* = 0.001)	Positive OFC is correlated with higher almond sIgE levels (*p* < 0.001) and higher peanut IgE levels (*p* = 0.003)	Not studied	95% predicted probability of a positive OFC is at SPT = 46 mm and almond-specific IgE = 174 kU/L	Almond-sIgE, almond SPT and age at challenge combined demonstrated good PV for Grade 2/3 allergic reactions
Chong et al., 2019, Singapore [79]	Retrospective	328: 269 AP, 59 TP	Peanuts	SPT ≥ 8 mm is highly predictive (>95%) of an allergic reaction	Peanut sIgE ≥ 6 kU/L provided a 95% PPV for clinical reaction	Not studied	Both cutoffs provided a 95% PPV for clinical reaction	These cutoff values can assist clinicians in assessing the risk of peanut OFC
Chua et al., 2021, China [80]	Single-center, cross-sectional with addition of retrospective data	31 (1–18 y): 16 AS, 15 TS	Peanuts	At 6 mm: ~95% specificity (*p* < 0.0001)	No differences in peanut sIgE between TS and AS	Ara h 2 sIgE 0.14 kU(A)/L = highest sensitivity and specificity; at 0.74 kU(A)/L, ~95% specificity (*p* = 0.02)	SPT (*p* < 0.0001) and Ara h 2 sIgE (*p* = 0.02) correlate with peanut allergy sIgE (*p* = 0.26), Ara h 1 sIgE (*p* = 0.19) and Ara h 3 sIgE (*p* = 0.27) do not correlate with peanut allergy	SPT results are the best predictor of peanut allergy in the Chinese pediatric population, followed by Ara h 2 sIgE
Lee et al., 2021, Korea [81]	Retrospective	377 (<3 y): 116 for egg whites, 182 for cow’s milk, 17 for walnuts, 22 for soybeans	Walnuts and others	Not studied	Walnut sIgE ≥ 0.35 kU/L: Class 1 = 9/10 (90.0%); Class 2 = 2/3 (66.7%); Class 3 = 1/4 (25.0%)	Not studied	Walnut sIgE ≥ 0.35 kU/L: PPV 90% in class 1 and 84.6% in classes 1–2 (*p* < 0.05); NPV 57.1% in class 1 and 75% in classes 1–2 (*p* < 0.05)	There is a correlation between clinical symptom profile and sIgE sensitization rate in patients with walnut allergy (accuracy rate of 76.5%)
Kubota et al., 2022, Japan [82]	Retrospective	41 (7.7 y median age); SG: 21; CG: 20	Macadamia nuts	Not studied	Median value of sIgE: in An group, 7.97 kUA/L; in non-An group, 1.92 kUA/L; in TS, 1.90 kUA/L	Not studied	sIgE median levels are higher in An group than in non-An group (*p* = 0.02) and CG (*p* < 0.001)	Optimal sIgE cutoff value is at 3.76 kUA/L, with sensitivity and NPV of 100%
Grinek et al., 2023, USA [83]	Retrospective	74 from LEAP cohort (from 4–11 m to 5 y)	Peanuts	Not studied	IgE and IgG4 against specific epitopes: sesIgE, Ara h sIgE, ses-IgG4	Not studied	Combining sesIgE and IgE to Ara h 1, 2, 3 and 9 could predict the peanut allergy status at 5 y (average validation accuracy of 64%)	IgE antibody profiles at 1 y of age can predict the outcome of peanut OFC at 5 y
Koutlas et al., 2023, USA [84]	Retrospective	510 (169 peanuts)	Peanuts (173 OFCs), tree nuts (85 OFC), others (eggs, milk)	Peanut OFC outcomes were not associated with median peanut SPT results	Peanut OFC outcome was not associated with median peanut sIgE levels	Not studied	Food sIgE level < 2 kU/L and SPT < 5 mm result in a valid cutoff with a failure rate of 13% for nonbaked milk, nonbaked eggs and nuts	SPT and sIgE are not predictive of the outcome of OFCs (*p* > 0.08)

m: months; y: years; SPT: skin prick test; OFC: oral food challenge; PV: predictive value; PPV: positive predictive value; NPV: negative predictive value; TS: tolerant subjects; AS: allergic subjects; AP: allergy to peanuts; TP: tolerance to peanuts; PPA: possible peanut allergy; FA: food allergy; SG: study group; CG: control group; An: anaphylaxis; LEAP: Learning Early About Peanut Allergy; sIgE: specific immunoglobulin E; tIgE: total immunoglobulin E; CI: confidence interval; sesIgE: sequential epitope-specific IgE.

## Data Availability

No new data were created or analyzed in this study.

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
