# Peer review of "Oral Food Challenge in Children with Tree Nut and Peanut Allergy: The Predictive Value of Diagnostic Tests"

_diagnostics, 2024, doi:10.3390/diagnostics14182069_

Round 1

Reviewer 1 Report

Comments and Suggestions for Authors

Interesting review dealing with the oral food challange in children with nut allergy-a potentially life threating condition.

Points of criticism:

1. For better understanding authors should show the flow chart for the diagnostic tests for nut allergy

2. Many patients with food allergy have multiple FA, could you shortly comment on the relevance of this phenomenon for the diagnosis of nut allergy. What about cross allergies ?

Author Response

RESPONSE TO REVIEWER n1

Point 1: For better understanding authors should show the flow chart for the diagnostic tests for nut allergy

Response 1: As you suggested, we have added a flow chart about diagnostic tests in nut allergy in section 3

Diagnostic tests. We have added “The main steps for the diagnosis of nut allergy are summarized in the

flowchart below (Fig. 1)” (lines 174-175) to introduce the figure of the flow chart.

We added the flow chart in the article as Figure 1, as reported below.

Fig. 1: Schematic flowchart of the main steps required for the diagnosis of nut allergies. NA: Nut Allergy; SPT: Skin Prick Test (lines 267-268)

Point 2: Many patients with food allergy have multiple FA, could you shortly comment on the relevance of this phenomenon for the diagnosis of nut allergy. What about cross allergies?

Response 2: Thanks for your suggestion. In lines 65-96 we have added a paragraph about multiple FA and

the phenomenon of cross reactions. We added “Considering the huge variety of foods related to nuts and peanuts, it is important to underline the cross-reactivity concept. It take places when a patient presents the allergic symptoms after the ingestion of closely related food [33]. However, a  real cross-reactivity among tree nut allergens is high between cashew and pistachio, as well as between walnuts and pecan [34]. Furthermore, in addition to a cross-reaction among nuts allergens, pollen sequence homologies are often present. This phenomenon is known as oral allergy syndrome (OAS), which occurs when an individual is initially sensitized to pollen and then presents a cross-reaction with a food. These reactions are caused by the so-called pan-allergens and are responsible of non-serious symptoms often localized to the oropharynx and mouth, which usually never result in anaphylaxis [35]. For example, the birch pollen Bet v1, is a ribosome-inactivating proteins (PR-10), one of the major pan-allergens in OAS. Walnut (Jug r5), hazelnut (Cor a1) and peanut (Ara h8), present homologous proteins, showing a cross-reactivity with Bet v1 [36]. Another pan-allergen is the peach lipid transfer protein, Pru p3, which is the primary sensitizing allergen for cross-reactivity with other lipid transfer proteins such as peanut (Ara h 9), hazelnut (Cor a 8), walnut (Jug r 3) and almond (Pru du 3) [37].”

We would like to correct the e-mail address of one of the authors. The correct email of Dr. Francesca Pastore is [email protected] (line 51). We apologize for the inconvenience.

Reviewer 2 Report

Comments and Suggestions for Authors

Review manuscript by Ludovica Cela et al describes current literature data on predictive values of SPT and sIgE levels on the outcome of OFC. Authors have gathered and described data on peanut and tree nut allergy. Discussion and conclusions of the presented findings could be elaborated further. In the current form this review provides description of the currently available literature.

Specific comments

Since this article analyses data on tree nut allergy and peanut allergy it should be reflected in the title. I suggest title as follows: Oral food challenge in children with tree nut and peanut allergy: the predictive value of diagnostic tests

Section 3. Diagnostic tests. First paragraph. STP wheal diameter should be SPT.

A cutoff of 0.1 kU/L should be discussed.

The diagnostic value of Ara h 2 should be discussed.

OAS abbreviation needs to be introduced.

The sentence about pecan protein contact eczema should be elaborated further.

Section methods and section discussion are both numbered 5. Please, correct.

Legend of the table 1 is labeled with table 2. Please, correct.

Comments on the Quality of English Language

Numerous mistakes in English language.  Please, pay attention.

Author Response

RESPONSE TO REVIEWER n2

Point 1: Since this article analyses data on tree nut allergy and peanut allergy it should be reflected in the title. I suggest title as follows: Oral food challenge in children with tree nut and peanut allergy: the predictive value of diagnostic tests

Response 1: As you suggested, we have changed the title in “Oral food challenge in children with tree-nut

and peanut allergy: the predictive value of diagnostic tests”.

Point 2: Section 3. Diagnostic tests. First paragraph. STP wheal diameter should be SPT.

Response 2: At line 104 we have changed STP in SPT, as you suggested.

Point 3: A cutoff of 0.1 kU/L should be discussed.

Response 3: The cut-off for sIgE positivity reported in our paper is 0.35 KU/L, following the EAACI

guidelines. However, the possibility of detection of sIgE is 0.1 KU/L.

Matricardi PM, EAACI Molecular Allergology User's Guide. Pediatr Allergy Immunol. 2016

May;27 Suppl 23:1-250. doi: 10.1111/pai.12563. PMID: 27288833.

Point 4: The diagnostic value of Ara h 2 should be discussed.

Response 4: Thanks for your suggestion. In lines 133-135 we have discussed better the diagnostic value of

Ara h 2, adding in the text “In a retrospective study conducted by Martinet et al. in 2016, they demonstrated that Ara h 2 had the best negative and positive predictive value. Moreover, the study showed that Ara h 2 titers can predict the risk of anaphylaxis [52].”.

Point 5: OAS abbreviation needs to be introduced.

Response 5: we added oral allergy syndrome (OAS) in line 76, so where you indicated (line 148) we left the abbreviation OAS.

Point 6: The sentence about pecan protein contact eczema should be elaborated further.

Response 6: As you suggested, we have elaborated further the sentence about pecan protein contact eczema, modifying in “Proteins resistant to high temperature and digestion are seemingly involved in severe systemic reactions to pecan, but protein contact eczema was also described, as in a case described in 2006 by Joyce et al., where a vesicular cutaneous reaction occurred in a patient after a prolonged contact with pecans, reminding that prolonged contact with allergenic food can lead to hypersensitivity reactions [57] [49].”  (lines 163-167).

Point 7: Section methods and section discussion are both numbered 5. Please, correct.

Response 7: In line 209 we have corrected the number of Discussion in 6

Point 8: Legend of the table 1 is labeled with table 2. Please, correct.

Response 8: In line 216 we have changed the title of the table in “Table 1”

We have corrected mistakes in the English language.

We would like to correct the e-mail address of one of the authors. The correct email of Dr. Francesca Pastore is [email protected] (line 51). We apologize for the inconvenience.